# Effects of Second Phases on Microstructure, Microhardness, and Corrosion Behavior of Mg-3Sn-(1Ca) Alloys

**DOI:** 10.3390/ma12162515

**Published:** 2019-08-07

**Authors:** Chunming Wang, Shuai Guo, Luming Zeng, Desen Zheng, Jianchao Xu, Munan Yang, Tongxiang Liang

**Affiliations:** 1School of Materials Science and Engineering, Jiangxi University of Science and Technology, Ganzhou 341000, China; 2Institute of Engineering Research, Jiangxi University of Science and Technology, Ganzhou 341000, China

**Keywords:** Mg-Sn-(Ca) alloys, second phase, microstructure, microhardness, corrosion behavior

## Abstract

The effects of second phases on microstructure, microhardness, and corrosion behavior of aged Mg-3Sn (T3) and Mg-3Sn-1Ca (TX31) alloys are investigated systematically. The thermal stability of the CaMgSn phase is higher than that of the Mg_2_Sn phase, and the microstructure remains essentially unchanged in the TX31 alloy after solution treatment for 28 h at 733 K. The T3 alloy exhibits double age-hardening peaks; one is 54.9 ± 2.1 HV for 7 h, and the other is 57.4 ± 2.8 HV for 15 h. However, the microhardness quickly reaches a stable value with increasing aging times in the TX31 alloy due to the no change in CaMgSn phases. It was also found by electrochemical impedance spectra that the corrosion resistance of aged T3 alloy is superior to that of aged TX31 alloy, especially T3 alloy aged for 7 h. The corrosion film of aged T3 alloy is denser, which attributes to most of dissolved Sn in the α-Mg matrix and the formation of a small quantity of tiny Mg_2_Sn particles, and effectively prevents the occurrence of further corrosion of the Mg matrix. However, galvanic cells formed between α-Mg and CaMgSn phases accelerate the corrosion of aged TX31 alloy.

## 1. Introduction

Magnesium (Mg) alloys, as the commercially lightest structural materials, have high strength/weight ratios, good damping capacity, and excellent thermal conductivity [1,2,3,4]. It has a wide range of applications in many fields, such as aerospace, transportation, automobile industries, and heat dissipation materials. However, limited formability and poor elevated temperature properties affect their process techniques [5]. The Sn element has lower diffusivity at high temperatures, appropriate solubility, and thermal-stability of the Mg_2_Sn phase in the Mg alloys. Thus, Mg-Sn based alloys are noteworthy materials for application at elevated temperatures [6,7,8,9]. In addition, Ca element forms a thermally stable CaMgSn phase (Sn/Ca weight ratio: 3/1~3.5/1) with Sn and Mg in the Mg-Sn alloy, which further improves the elevated temperature properties of the alloy [10,11,12,13]. However, the micromechanical properties and corrosion behavior of Mg-Sn alloys are rarely reported. The study of the micromechanical properties mainly focuses on the variation mechanism of residual stress and fracture toughness of Mg alloy. Ast and Ghidelli et al. [14,15] reviewed the micro-pillar splitting and micro-cantilever bending approaches to fracture toughness evaluation at the micro-scale, which can be used as a reference in the research of Mg alloy. 

For corrosion behavior of Mg-Sn alloys, Sn, Ca are all the commonly used elements for Mg corrosion, which was found in the pursuit of nontoxic metallic elements [16,17,18,19,20]. Research into the corrosion behavior of Mg-Sn [18,19,20,21,22,23,24,25] and Mg-Sn-Ca [26,27,28,29] alloys indicates that the corrosion resistance depends on the distribution of Sn, Ca elements, and the microstructure. Cao et al. [18] and Liu et al. [19] researched the corrosion behavior of Mg-Sn alloys by solution heat treatment. The corrosion resistance is improved with the increasing Sn concentration in Mg matrix, and the corrosion rate decreased in the alloy. In addition, the properties of the surface films changed with Sn addition, and the localized breakdown increased with a Sn increase [20,21,22]. It was also found that the corrosion resistance mainly relied on Mg_2_Sn precipitates, which functioned as pitting corrosion and markedly increased the hydrogen evolution rate [9,23,24]. For Mg-Sn-Ca alloys, the corrosion behavior of as-extruded Mg-Sn-Ca alloys was investigated and was influenced by the precipitated phases, such as CaMgSn or Mg_2_Sn and Mg_2_Ca, of the alloys according to the different Sn/Ca weight ratios [25,26]. Liu et al. [27] researched the effect of only a little Ca addition on corrosion behavior of Mg-8Sn-1Al-1Zn-Ca alloys and found that the refined grain size and the dispersed distribution of CaMgSn and Mg_2_Sn precipitates can improve the corrosion resistance of the alloy. Yang et al. [28] found that minor Ce has an effect on the corrosion behavior of Mg-3Sn-1Ca alloy. The addition of Ce can accelerate the formation of a denser corrosion product film. 

In the paper, based on Mg-3Sn (T3) alloy with a Mg_2_Sn phase and Mg-3Sn-1Ca (TX31) alloy with a CaMgSn phase, the effects of the single second phase on microstructure, microhardness, and corrosion behavior of T3 and TX31 alloys are investigated after solution and aging treatment. The microstructure and microhardness the two alloys are analyzed by X-ray diffraction, optical microscopy/scanning electron microscopy, and Vickers. The corrosion experiment is observed by an electrochemical workstation. Meanwhile, the relationship between microstructure and corrosion behavior, that is the corrosion mechanism of the two alloys, is also discussed.

## 2. Materials and Methods 

The Mg-3wt.% Sn and Mg-3wt.% Sn-1wt.% Ca alloys were prepared by a crucible resistance furnace, in which Ca was added in the form of Mg-30 wt.% Ca master alloy. The Mg ingots were firstly heated to 1023 K and melted sufficiently, the CO_2_+SF_6_ mixed gases were pumped into the furnace to protect the alloy from oxidation. The Sn and Mg-Ca master alloys were then added to the molten Mg metal and held for 30 min. To ensure homogeneity of the alloy elements, the molten metal was stirred and then sat for 5 min. The molten metal was finally cast in the steel molds (20 mm × 110 mm × 140 mm) preheated to 513 K. The specimens were cut to 10.0 mm × 10.0 mm × 3.0 mm. The cut samples were solution treated and isothermally aged at 733 K for 28 h and at 513 K for different times, respectively.

The microstructural characterization for T3 and TX31 samples was observed by Primo star (Carl Zeiss, Heidenheim, Germany) optical microscopy (OM) and JEOL-JSM-6490LV (Tokyo, Japan) scanning electron microscopy (SEM) equipped with Noran Vantage DS (Thermo Fisher Scientific, Waltham, MA, USA) Energy Dispersive X-ray (EDS). Four volume percent nitric acid and alcohol were used as an etching solution. The phase structure analysis of the T3 and TX31 alloys was examined by Parnike X’ Pert PRO MPD (Holland Panalytical, Almelo, Netherlands) X-ray diffraction (XRD) with Cu Kα radiation. The microhardness was obtained using the 200HVS-5 Vickers under a load of 25 g.

The corrosion behavior of the T3 and TX31 alloys with different heat treatments was tested by an electrochemical workstation (CHI760E, Shanghai Chenhua, Shanghai, China). The standard three-electrode cell used includes a reference electrode (saturated calomel electrode), a working electrode (samples), and a counter electrode (platinum). The test area of the samples was an exposed surface area of 10 mm × 10 mm embedded into epoxy resin. The electrolyte was 3.5 wt.% NaCl solution and was used at room temperature (about 303 K). For electrochemical impedance spectroscopy (EIS), the scan rate of the test samples was 5 mV·s^−1^. The frequency range of EIS was from 100 kHz to 0.01 Hz with a perturbation amplitude of 5 mV, and the EIS data were fitted by Z view software (ZsimDemo3.30d).

## 3. Results and Discussion

Figure 1 shows the XRD patterns of T3 and TX31 alloys before and after solid solution treatment. α-Mg and Mg_2_Sn phases existed in the as-cast T3 alloy. The intensities of Mg_2_Sn peaks disappeared, and the T3 alloy had only an α-Mg phase after solid solution treatment for 28 h at 733 K, suggesting that the Mg_2_Sn phases were decomposed and the Sn elements were dissolved into the α-Mg matrix in the process of solid solution treatment. For the TX31 alloy, the α-Mg phase and CaMgSn phase were found in the as-cast alloy. Compared with the intensities of the CaMgSn phase in the as-cast alloy, the intensities CaMgSn peaks hardly changed in the solution treated TX31 alloy. It was demonstrated that the thermal stability of CaMgSn phase was higher compared with Mg_2_Sn phase. 

The OM morphologies of the as-cast T3 (a) and TX31 (b), solution treated T3 (c) and TX31 (d) alloys for 28 h at 733 K are shown in Figure 2. The as-cast T3 alloy exhibited a typical dendritic structure, and the segregation of Sn elements occurred in the α-Mg matrix (Figure 2a). The dendritic structure of the alloy disappeared after solid solution treatment for 28 h at 733 K, suggesting that the Mg_2_Sn phases were decomposed, as shown in Figure 2c. For the as-cast TX31 alloy, the rod-like and small granular CaMgSn phases are observed in Figure 2b. These phases were distributed uniformly in the α-Mg matrix. It was also found that the microstructure remained essentially unchanged in the solution treated TX31 alloy for 28 h at 733 K compared with the as-cast TX31 alloy (Figure 2d), which is consistent with the XRD results in Figure 1.

Figure 3 shows the XRD patterns of T3 (a) and TX31 (b) alloys with different aging times at 513 K. According to the indexed results, the intensities of Mg_2_Sn precipitates increased with an increase in aging times, indicating that the number of Mg_2_Sn precipitates increased in the aged T3 alloy (Figure 3a). However, the intensities of CaMgSn phases hardly changed with the prolongation of aging time in the aged TX31 alloy (Figure 3b), which is attributed to high thermal stability of the CaMgSn phase.

Figure 4 shows the microhardness of aged T3 and TX31 alloys at 513 K for different aging times. The microhardness of the T3 and TX31 alloys were 40.1 ± 2.1 and 47.0 ± 2.1 HV after solid solution treatment, respectively. Compared with aged T3 alloy, the microhardness of aged TX31 alloy was higher due to the existence of CaMgSn phase after solid solution treatment, as shown in Figure 1 and Figure 2d. As the aging time increased, the microhardness of T3 alloy increased to a local maximum value of 54.9 ± 2.1 HV at 7 h. Then, it was found that the microhardness decreased slightly with an increase in the aging times until it reached the minimum value at 11 h. After that, the microhardness increased again to reach a peak value of 57.4 ± 2.8 HV at 15 h. It then began to decrease slowly to about 50 HV after aging for 22 h (Figure 4a). Combined with the XRD results in Figure 3a, the aging hardness was associated with the Mg_2_Sn precipitates, which can lead to the double-peak aging behavior. For TX31 alloy, the microhardness at first increased and then quickly reached a stable value of about 49.5 HV with increasing aging times, which is attributed to the no change of CaMgSn phases in the TX31 alloy.

Figure 5 shows the EIS analysis of aged T3 and TX31 alloys in 3.5 wt.% NaCl solution. For the T3 alloy with different aging times, the capacitive loops are shown in Figure 5a in the frequency range from 1 Hz to 10^5^ Hz. The T3 alloy with 0 h and 7 h aging had a higher global impedance compared to that of T3 alloy with 15 h and 30 h aging. The global impedance values of the alloy reached the maximum at 7 h aging. For the TX31 alloy with different aging times, the global impedance values for 7 and 15 h aging were higher compared to the solution treated alloy (Figure 5c). Meanwhile, the global impedance values for 7 and 15 h aging showed no significant difference. It indicates that the T3 and TX31 alloys with 7 h aging increased barrier properties and inhibited the occurrence of corrosion. 

The bode plots of the two kinds of alloys with different aging time are shown in Figure 5b,d, the impedance modulus (|Z|) of the aged T3 alloy for 7 h was higher than that of the other aged alloys in the low-frequency range from 1 to 10 Hz. Moreover, the impedance modulus (|Z|) of the aged TX31 alloy for 7 h and 15 h was basically the same, and greater than that of the solution treated alloy. It also suggests that the two kinds of aged alloys for 7 h have better corrosion resistance. The phase angle plots that show a capacitive behavior over the frequency range reflect the surface state and structural changes of the alloy. The native passivation layer on Mg surface was characterized at high and mid frequencies (10^5^~1), and the passivation layer was affected by surface state and structural changes in the alloy [29]. Figure 5b,d show the phase angle plots of the alloys. The values of the phase angle of aged T3 alloy for 7 h were more negative at the frequency range of 1 Hz–10^2^ Hz. For aged TX31 alloy for 7 h, the phase angle is always more negative at the frequency range of 1 Hz–10^4^ Hz, indicating that the passive films can effectively restrain corrosion. However, the phase angle of aged T3 alloy for 15 h was more negative at the frequency ranges of 10^3^ Hz–10^5^ Hz, suggesting the properties of corrosion films may change. In addition, the global impedance and |Z| of the aged T3 alloy were higher than that of the aged TX31 alloy (Figure 5a–d), suggesting that the aged T3 alloy has better corrosion resistance.

According to the equivalent circuit of the alloys in Figure 5e [30], the fitting values of EIS spectra are summarized in Table 1. R_sol_ is the solution resistance between the working and reference electrodes, R_t_ and CPE_1_ are the resistance and constant phase of corrosion product film, respectively, R_f_ and CPE_2_ are the charge transfer resistance and capacitance. The R_sol_ of the two kinds of alloys remained basically unchanged. The R_t_ and R_f_ of the aged T3 alloy were higher compared with the aged TX31 alloy, which indicates that the film formed on the aged T3 alloy is more protective than that of the aged TX31 alloy [31].

To further understand the difference in corrosion mechanisms in the aged T3 and TX31 alloys, the potentiodynamic polarization curves and Nyquist plots in the frequency range from 10 mHz to 100 kHz of T3 and TX31 alloys for 7 h at 513 K are shown in Figure 6. According to the potentiodynamic polarization curves in Figure 6a, the corrosion potential (E_corr_, V), corrosion current density (i_corr_, A·cm^−2^), Tafel slope (β_a_, β_c_, V/decade of current density) and polarization resistance (R_pol_, Ω·cm^2^) are obtained in Table 2. The β_a_, β_c_, and R_pol_ were calculated by the Tafel extrapolation method and the Stern-Geary formula, respectively. The stern-Geary formula is as follows: (1)$$R_{pol}=\frac{\beta_{a}\times \beta_{c}}{2.303\times i_{corr}\times (\beta_{a}+\beta_{c})}.$$

The E_corr_ and i_corr_ of the aged T3 alloy were −1.559 V and 1.47 × 10^−5^ A·cm^−2^, respectively. Compared to the aged TX31 alloy, the value of i_corr_ decreased, indicating that the corrosion resistance was improved because the corrosion rate can be accurately reflected by the i_corr_ value. Meanwhile, the R_pol_ (1633.21 Ω·cm^2^) of the aged T3 alloy was higher, which is consistent with the EIS results in Figure 5 and Table 1. The aged T3 alloy contained a high capacitive loop at high frequencies (HF, 100 kHz–1 Hz), a low capacitive loop at medium frequencies (MF, 1Hz–26.1 mHz), and an inductive loop at low frequencies (LF, 26.1 mHz–10 mHz), as shown in Figure 6b. However, the MF capacitive loop was not very obvious in the aged TX31 alloy. The HF and MF capacitive loops are usually attributed to both charge transfer and surface film effect, and the relaxation of mass transport in the growing oxide, respectively [18,32,33]. Combined with the above analysis, it was found that the corrosion mechanisms are different in the aged T3 and TX31 alloys.

The SEM morphologies and EDS spectra of aged T3 and TX31 alloys for 7 h at 513 K are shown in Figure 7. Table 3 shows the element contents (1–8 points in Figure 7a,b) of aged T3 (a) and TX31 (b) by EDS analysis. For aged T3 alloy, it can be observed from points 1, 2, and 3 in Figure 7a that the Sn elements were abundant in the α-Mg matrix. Combined with the XRD results in Figure 3a, the precipitated Mg_2_Sn is not clearly observed in Figure 7a due to the small amount of nano-sized precipitates, which is in good agreement with the previous research [4,34]. There are some larger Mg_2_Sn particles (4 point and Red circles) in Figure 7a, which may be due to the residues after solid solution treatment. For aged TX31 alloy, the solute Sn, Ca atoms were not found in points 5, 6, and 7 of the α-Mg matrix (Figure 7b). According to the XRD results (Figure 3b) and the EDS of point 8, it was found that the rod shape and small granular particles were the CaMgSn phases in Figure 7b. The Sn, Ca elements existed mainly in the form of CaMgSn phases, which means the CaMgSn phase was hardly decomposed in the α-Mg matrix after heat treatment.

Figure 8 shows the surface morphologies (a–f) and EDS spectra (g,h) of aged T3 and TX31 alloys for 7 h at 513 K after immersion in 3.5 wt.% NaCl solution for 12 h. To improve the electric conductivity of the corrosion film and make the corrosion film easier to observe, the surface of the alloys was coated with an ultrathin layer of platinum (Pt). The aged TX31 alloy was more severely corroded because the galvanic cells formed between α-Mg and a large number of CaMgSn phases accelerate the corrosion of the alloy (Figure 7b and Table 3). Meanwhile, some pit etchings (Red circles) are also found in Figure 8b. Compared with the aged TX31 alloy, there were no obvious pit etchings in the aged T3 alloy (Figure 8a).

The corrosion product film of aged T3 and TX31 alloys are shown in Figure 8c–f. A honeycomb corrosion microstructure was composed of intertwined nanoplates [28,35]. These nanoplates were distributed uniformly on the surface of the alloy. Meanwhile, it was found that some nanoclusters were formed by nanoplates on the nanosheet layer, which could supplement some pores created by the nanoplate layer and prevent further corrosion of the alloys. But some cracks were also found on the surface film of aged TX31 alloy (Figure 8d) due to the existence of CaMgSn phase formed the Ca, Sn and Mg elements, which may be the main cause of corrosion deterioration in the alloy. In addition, the nanosheets of aged T3 alloy were smaller and interlaced more densely compared with the TX31 alloy (Figure 8e,f). The denser corrosion film of aged T3 alloy is attributed to the dissolved Sn and the formation of a small quantity of nano-sized Mg_2_Sn (Figure 7 and Table 3). The area g in (c) was analyzed by EDS spectra, the corrosion film of aged T3 alloy contains Sn expect Mg and O compared with aged TX31 alloy.

Figure 9 shows XRD patterns of aged T3 and TX31 alloys for 7 h at 513 k after immersion in 3.5 wt.% NaCl solution for 12 h. Compared with the XRD results of Figure 3, the two alloys all contained Mg(OH)_2_ phases, suggesting that the corrosion product after immersion is mainly Mg(OH)_2_. It was also found that the intensities of Mg(OH)_2_ phase of the TX31 alloy were higher than that of T3 alloy. Combined with the EDS results in Figure 8g,h, the content of Mg and Sn elements decreased and increased, respectively. The presence of Sn indicates the corrosion product has tin oxides (SnO_2_) in the T3 alloy. The formation of tin oxides is mainly attributed to the residual Sn atoms in the α-Mg matrix (1–3 points in Figure 7a). The SnO_2_ in the corrosion products can inhibit the formation of Mg(OH)_2_ and enhance the compatibility of corrosion product film because the Pilling–Bedworth Ratio of stannic oxide is 1.32 [19], so the corrosion resistance of aged T3 alloy was better because of a significant number of dissolved Sn atoms and a small quantity of tiny Mg_2_Sn particles.

## 4. Conclusions

In this paper, the effects of second phases on microstructure, microhardness, and corrosion behavior of aged Mg-3Sn and Mg-3Sn-1Ca alloys were investigated systematically at 513 K for different times. The following conclusions are drawn from this study:(1)The Mg_2_Sn phases of Mg-3Sn alloy were decomposed in the α-Mg matrix, and the microstructure of Mg-3Sn-1Ca alloy remained essentially unchanged after solution treatment for 28 h at 733 K, suggesting that the thermal stability of the CaMgSn phase is evidently higher compared to that of the Mg_2_Sn phase.(2)The Mg-3Sn alloy exhibited double age-hardening peaks. The microhardness increased to a first peak value of 54.9 ± 2.1 HV at 7 h with the increase in aging times. After a slight decrease, the microhardness increased again to a second peak value of 57.4 ± 2.8 HV for 15 h. The microhardness of aged Mg-3Sn-1Ca alloy quickly reached a stable value with the increasing aging times due to the no change of CaMgSn phases.(3)The corrosion resistance of aged Mg-3Sn alloy was obviously better compared to aged Mg-3Sn-1Ca alloy, especially Mg-3Sn alloy aged for 7 h. The most of dissolved Sn atoms and a small quantity of tiny Mg_2_Sn particles made the corrosion film composed of SnO_2_ become denser, and inhibited further corrosion of the alloy. However, a significant number of galvanic cells formed between α-Mg and a large number of CaMgSn phases accelerated the corrosion of aged Mg-3Sn-1Ca alloy.

## Figures and Tables

**Figure 1 materials-12-02515-f001:**
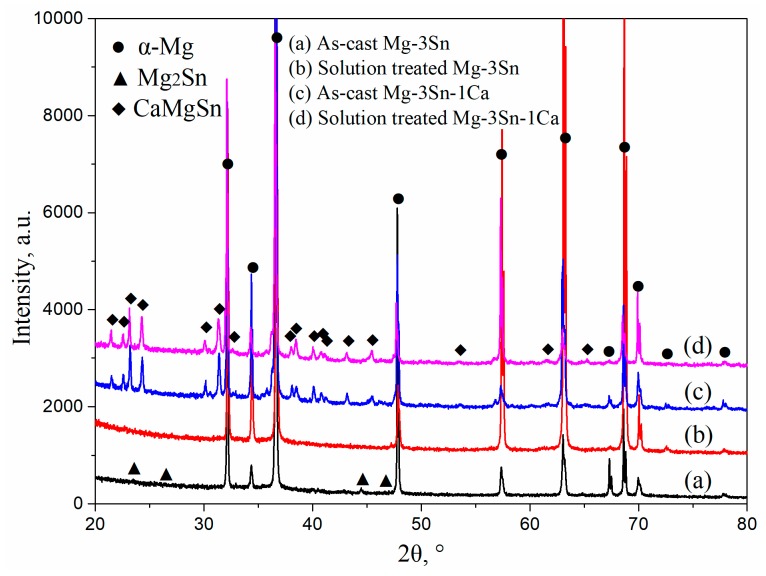
XRD patterns of as-cast (**a**,**c**) and solution treated (**b**,**d**) Mg-3Sn and Mg-3Sn-1Ca alloys.

**Figure 2 materials-12-02515-f002:**
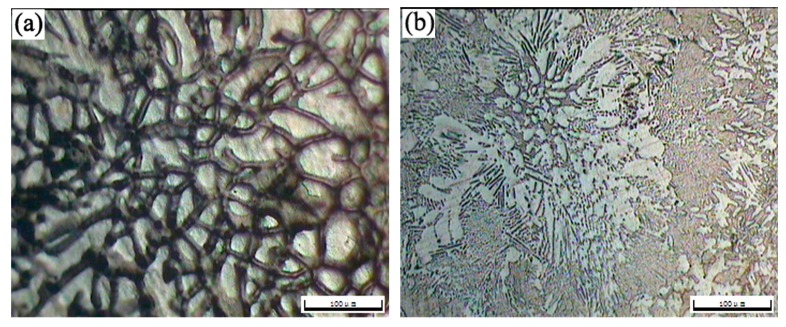
OM images of as-cast Mg-3Sn (**a**) and Mg-3Sn-1Ca (**b**), solution treated Mg-3Sn (**c**), and Mg-3Sn-1Ca (**d**) alloys for 28 h at 733 K.

**Figure 3 materials-12-02515-f003:**
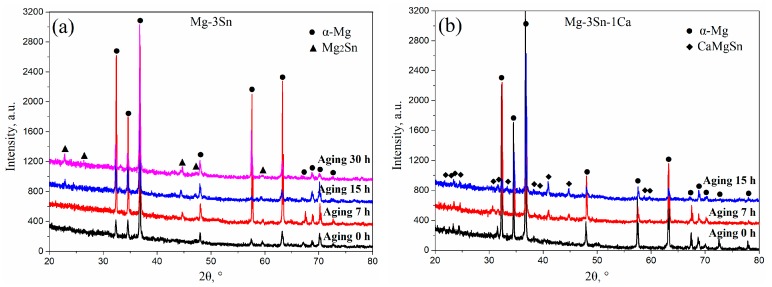
XRD patterns of aged Mg-3Sn (**a**) and Mg-3Sn-1Ca (**b**) alloys with different aging times at 513 K.

**Figure 4 materials-12-02515-f004:**
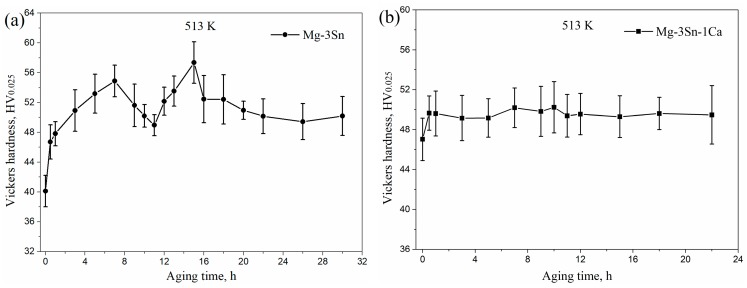
The microhardness of Mg-3Sn (**a**) and Mg-3Sn-1Ca (**b**) alloys with different aging times at 513 K.

**Figure 5 materials-12-02515-f005:**
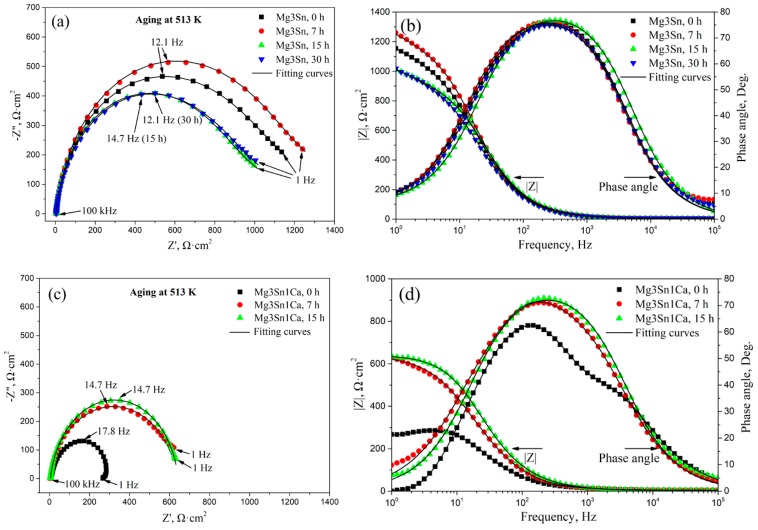
EIS analysis of aged Mg-3Sn and Mg-3Sn-1Ca alloys in 3.5 wt.% NaCl solution: (**a**,**c**) Nyquist plot, (**b**,**d**) Bode and Phase angle plots; The equivalent circuit (**e**) of aged Mg-3Sn and Mg-3Sn-1Ca alloys.

**Figure 6 materials-12-02515-f006:**
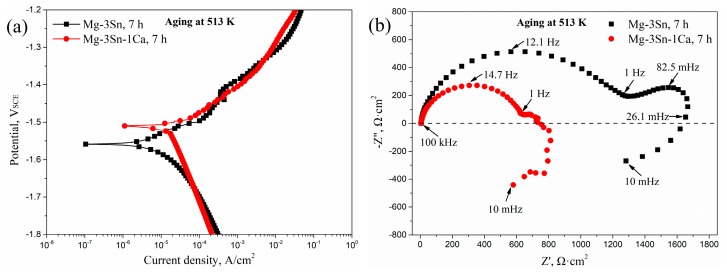
Potentiodynamic polarization curves (**a**) and Nyquist plot (**b**) of Mg-3Sn and Mg-3Sn-1Ca alloys for 7 h at 513 k.

**Figure 7 materials-12-02515-f007:**
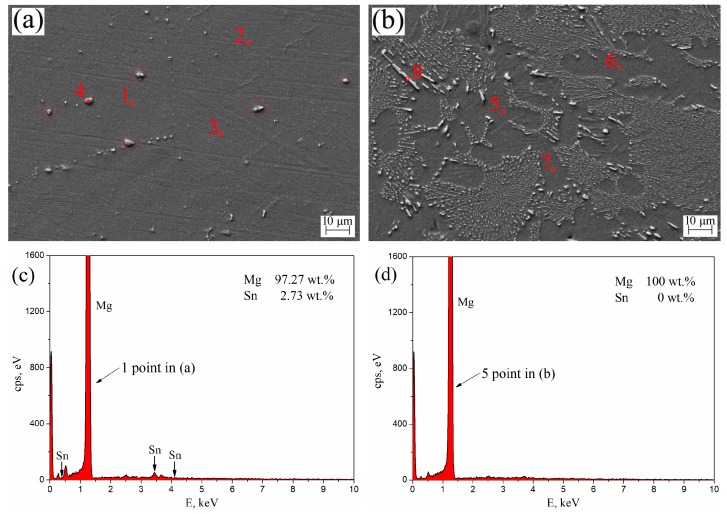
SEM morphologies of aged Mg-3Sn (**a**) and Mg-3Sn-1Ca (**b**) alloys for 7 h at 513 K; (**c**) EDS spectra 1 point in (**a**); (**d**) EDS spectra 5 point in (**b**).

**Figure 8 materials-12-02515-f008:**
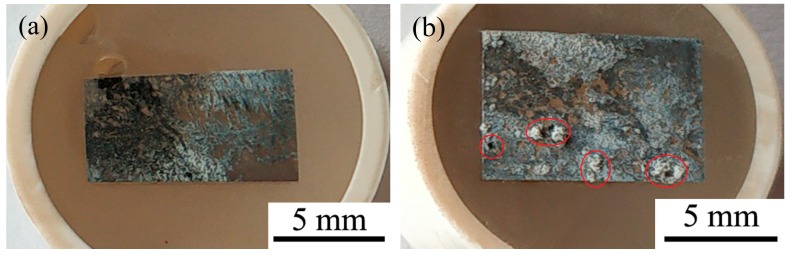
Surface morphologies (**a**–**f**) and EDS spectra (**g**,**h**) of aged Mg-3Sn and Mg-3Sn-1Ca alloys for 7 h at 513 k after immersion in 3.5 wt.% NaCl solution for 12h: (**a**,**c**,**e**) Mg-3Sn; (**b**,**d**,**f**) Mg-3Sn-1Ca; (**g**) Area g in (**c**); (**h**) Area h in (**d**).

**Figure 9 materials-12-02515-f009:**
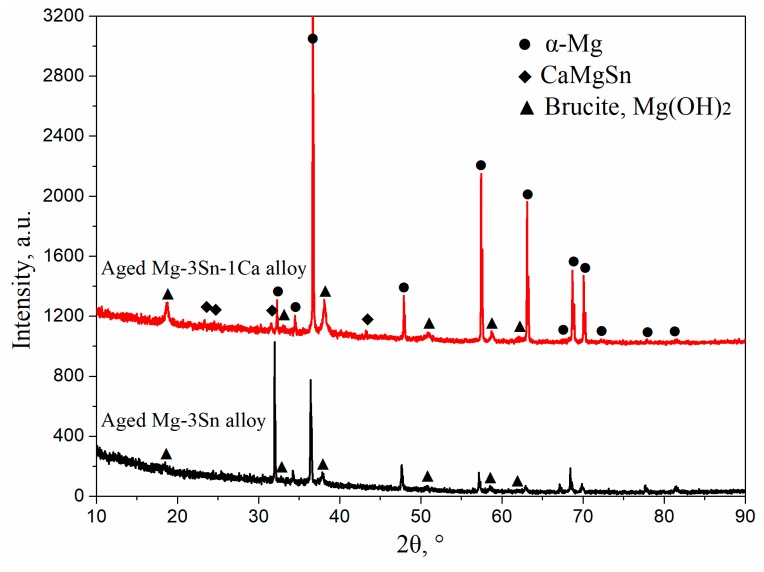
XRD patterns of aged Mg-3Sn and Mg-3Sn-1Ca alloys for 7 h at 513 K after immersion in 3.5 wt.% NaCl solution for 12 h.

**Table 1 materials-12-02515-t001:** Fitting parameters obtained from EIS spectra of aged Mg-3Sn and Mg-3Sn-1Ca alloys in 3.5 wt.% NaCl solution.

Specimen	R_sol_ (Ω·cm^2^)	CPE_1_ (F/cm^2^)	n_1_	R_t_ (Ω·cm^2^)	CPE_2_ (F/cm^2^)	n_2_	R_f_ (Ω·cm^2^)
T3, 0 h	4.931	1.543 × 10^−5^	0.9322	950.9	0.934 × 10^−3^	0.5128	480.6
T3, 7 h	5.463	1.572 × 10^−5^	0.9246	1123	1.284 × 10^−3^	0.5877	425.9
T3, 15 h	4.236	1.340 × 10^−5^	0.9398	848.4	1.200 × 10^−3^	0.5367	405.0
T3, 30 h	4.841	1.708 × 10^−5^	0.9242	863.4	1.135 × 10^−3^	0.5637	373.1
TX31, 0 h	4.598	2.039 × 10^−5^	0.8984	290.2	1.473 × 10^−5^	0.9635	21.13
TX31, 7 h	4.577	3.191 × 10^−5^	0.8893	693.6	2.993 × 10^−5^	0.9877	46.67
TX31, 15 h	4.486	0.716 × 10^−5^	0.9609	636.9	1.625 × 10^−5^	0.8751	18.60

**Table 2 materials-12-02515-t002:** Tafel fitting results of Mg-3Sn and Mg-3Sn-1Ca alloys for 7 h at 513 K in 3.5 wt.% NaCl solution.

Alloy	E_corr_ (V/SCE)	i_corr_ (A·cm^−2^)	β_c_ (V/decade)	β_a_ (V/decade)	R_pol_ (Ω·cm^2^)
T3, Aging 7 h	−1.559	1.47 × 10^−5^	174.57 × 10^−3^	80.92 × 10^−3^	1633.21
TX31, Aging 7 h	−1.510	1.91 × 10^−5^	252.46 × 10^−3^	75.18 × 10^−3^	1316.95

**Table 3 materials-12-02515-t003:** Element contents of aged Mg-3Sn (**a**) and Mg-3Sn-1Ca (**b**) by EDS analysis in Figure 7.

Element	1	2	3	4	5	6	7	8
Mg, wt.%	97.27	96.88	96.72	82.42	100	100	100	94.14
Sn, wt.%	2.73	3.12	3.28	17.58	0	0	0	4.16
Ca, wt.%	-	-	-	-	0	0	0	1.70

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
