# Peer review of "Effects of Second Phases on Microstructure, Microhardness, and Corrosion Behavior of Mg-3Sn-(1Ca) Alloys"

_materials, 2019, doi:10.3390/ma12162515_

Round 1

Reviewer 1 Report

Comments to the Author:
The authors of this paper present a systematic investigation of the effects of second phases on microstructure, microhardness and corrosion behavior of aged Mg-3Sn and Mg-3Sn-1Ca alloys. Nevertheless, some details should be considered by the authors:

Introduction

COMMENT: Page 1, lines 58-62: The aim of the manuscript, which is presented at the end introduction, should be more descriptive including all the analysis carried out.

Materials and Methods: All the methods are described in detail.

Results and discussion

COMMENT: Page 3, 4 and 9, Figs 1, 3 and 9: I suggest the authors to add a quantitative analysis of the relative intensities of the peaks that emerge in the XRD patterns.

COMMENT: Page 5, line 159: Ι suggest the authors add more specific details about the surface state and structural changes of the alloy.

To conclude, reported data are sufficiently discussed and commented and the results support sufficiently the authors conclusion. Therefore, I think that this paper is suitable for publication on Journal of Porous Materials.

Author Response

Dear Reviewer,

Thank you very much for your careful review and comments on our manuscript. These suggestions will definitely improve the quality of our manuscript. We have revised the manuscript according to your comments:

1. Page 1, lines 58-62: The aim of the manuscript, which is presented at the end introduction, should be more descriptive including all the analysis carried out.

Response: Thanks for your reminding. The aim of the manuscript have been added the descriptive analysis. The revision has been marked by red in the revised manuscript.  

2. Page 3, 4 and 9, Figs 1, 3 and 9: I suggest the authors to add a quantitative analysis of the relative intensities of the peaks that emerge in the XRD patterns.

Response: Thanks for your comments. According to the Rietveld refinement of XRD, it is found that the quantitative analysis does not yield results because the number of the second phase is too small beyond the range of the XRD accuracy. We have modified the XRD part by the intensities of phase peaks. The revision has been marked by red in the revised manuscript. 

3. Page 5, line 159: Ι suggest the authors add more specific details about the surface state and structural changes of the alloy.

Response: Thanks for your comments. we have added the related details about the surface stat and structural changes of the alloy. The revision has been marked by red in the revised manuscript. 

Reviewer 2 Report

The authors provide a paper dealing with the effects of second phases on microstructure, microhardness and corrosion behavior of Mg-3Sn-(1Ca) alloys. The paper can be of interest for Materials, nevertheless MAJOR revisions are requested.

I would like that the authors comment on the role of residual stress and fracture toughness on the mechanical behavior. The authors must be aware that there exist techniques to extract both quantities at the micrometer scale see doi.org/10.1016/j.matdes.2019.107762 and doi.org/10.1016/j.matdes.2016.06.003, for instance. The authors are invited to include these works in the introduction while commenting on these two quantities based on the literature. This is a very relevant topic for the present paper dealing with micromechanical properties of Mg alloys. I would like that the authors improve the nanoindentation analysis. Specifically, it does not seem to me that there is any trend as a function of the aging behavior. The authors must comment more on that linking the data with the alloy microstructural evolution. Moreover, the authors should comment about the indents providing some SEM images. I would like that the authors improve the EDX analysis in Figure 7. The authors must provide better quality of graphs as well as a sample mapping showing the compositional variations.

Author Response

Dear Reviewer,

Thank you very much for your careful review and comments on our manuscript. These suggestions will definitely improve the quality of our manuscript. We have revised the manuscript according to your comments:

1. I would like that the authors comment on the role of residual stress and fracture toughness on the mechanical behavior. The authors must be aware that there exist techniques to extract both quantities at the micrometer scale see doi.org/10.1016/j.matdes.2019.107762 and doi.org/10.1016/j.matdes.2016.06.003, for instance. The authors are invited to include these works in the introduction while commenting on these two quantities based on the literature. This is a very relevant topic for the present paper dealing with micromechanical properties of Mg alloys.

Response: Thanks for your reminding. These works about residual stress and fracture toughness on the mechanical behavior are included the introduction, and the revision has been marked by red in the revised manuscript.  

2. I would like that the authors improve the nanoindentation analysis. Specifically, it does not seem to me that there is any trend as a function of the aging behavior. The authors must comment more on that linking the data with the alloy microstructural evolution. Moreover, the authors should comment about the indents providing some SEM images.

Response: Thanks for your comments. In the paper, on the basis of T3 alloy with Mg2Sn phase and TX31 alloys with CaMgSn phase, the effects of single second phase on microstructure, microhardness and corrosion behavior of T3 and TX31 alloys are investigated after solution and aging treatment. Meanwhile, the relationship between microstructure and corrosion behavior, that is, the corrosion mechanism of the two alloys is also discussed. There is no focus on micromechanical properties (residual stress and fracture toughness) in the paper. But the expert have gave us a good idea and research direction by the references (doi.org/10.1016/j.matdes.2019.107762 and doi.org/10.1016/j.matdes.2016.06.003). We hope to make better progress in the future research.

3. I would like that the authors improve the EDX analysis in Figure 7. The authors must provide better quality of graphs as well as a sample mapping showing the compositional variations.

Response: Thanks for your comments. According to your suggestion, we have revised and improved the EDX analysis in Figure 7. Meanwhile, the compositional variations of graphs are well observed in the EDS mapping, as shown in Figure 7.

Thank you and best regards.

Sincerely yours

Chunming Wang

Reviewer 3 Report

1) In my opinion, the word "erosion" is not used correctly in the manuscript.

Erosion is a degradation of metal due to mechanical action (friction, abrasion e.q. by a flowing liquid, or impinging by particles suspended in flowing liquid or gas) and erosion usually occurs with corrosion.

In this work there was no mechanical action of the corrosion environment on the samples. So the word "erosion" should be changed into the word "corrosion" (lines: 26, 162, 227, 243, 281).

2) line 81: Please specify the temperature of the electrolyte during electrochemical tests.

3) Please complete missing marks and values on the Y-axis on the XRD patterns and EDS spectra in Fig.1, Fig.3a)-b), Fig.7c)-d), Fig.8g)-h) and Fig.9.

Author Response

Dear Reviewer,

Thank you very much for your careful review and comments on our manuscript. These suggestions will definitely improve the quality of our manuscript. We have revised the manuscript according to your comments:

1. In my opinion, the word "erosion" is not used correctly in the manuscript. Erosion is a degradation of metal due to mechanical action (friction, abrasion e.q. by a flowing liquid, or impinging by particles suspended in flowing liquid or gas) and erosion usually occurs with corrosion. In this work there was no mechanical action of the corrosion environment on the samples. So the word "erosion" should be changed into the word "corrosion" (lines: 26, 162, 227, 243, 281).

Response: Thanks for your reminding. We have revised the related content about the word "erosion" and replaced with the word "corrosion", and the revision has been marked by red in the revised manuscript.

2. line 81: Please specify the temperature of the electrolyte during electrochemical tests.

Response: Thanks for your reminding. We have specified that the electrolyte was used at room temperature (about 303 K). And the revision has been marked by red in the revised manuscript.

3. Please complete missing marks and values on the Y-axis on the XRD patterns and EDS spectra in Fig.1, Fig.3a)-b), Fig.7c)-d), Fig.8g)-h) and Fig.9.

Response: Thanks for your reminding. We have marked the Y-axis values on the XRD patterns and EDS spectra in Fig.1, Fig.3a)-b), Fig.7c)-d), Fig.8g)-h) and Fig.9.

Thank you and best regards.

Sincerely yours,

Chunming Wang

Round 2

Reviewer 2 Report

-